# Prognostic Significance of the Cribriform Pattern in Prostate Cancer: Clinical Outcomes and Genomic Alterations

**DOI:** 10.3390/cancers16071248

**Published:** 2024-03-22

**Authors:** Mutlay Sayan, Yetkin Tuac, Mahmut Akgul, Grace K. Pratt, Mary D. Rowan, Dilara Akbulut, Samet Kucukcolak, Elza Tjio, Shalini Moningi, Jonathan E. Leeman, Peter F. Orio, Paul L. Nguyen, Anthony V. D’Amico, Cagdas Aktan

**Affiliations:** 1Department of Radiation Oncology, Dana Farber Cancer Institute, Brigham and Women’s Hospital, Harvard Medical School, Boston, MA 02215, USA; 2Department of Statistics, Ankara University, 06100 Ankara, Türkiye; yetkin.tuac@ankara.edu.tr; 3Department of Pathology and Laboratory Medicine, Albany Medical Center, Albany, NY 12208, USA; 4Center for Cancer Research, Laboratory of Pathology, National Institutes of Health, Bethesda, MD 20892, USA; 5Department of Pathology and Laboratory Medicine, Rutgers University, New Brunswick, NJ 08901, USA; 6Histopathology Department, Harrogate District Hospital, Harrogate HG2 7SX, UK; 7Department of Medical Biology, Faculty of Medicine, Bandirma Onyedi Eylul University, 10250 Balikesir, Türkiye

**Keywords:** prostate cancer, cribriform pattern, genomic alterations, radical prostatectomy, progression-free survival

## Abstract

**Simple Summary:**

Prostate cancer is a condition with varying outcomes, and understanding its progression is vital for treatment. Our research focused on a specific pattern observed in tumor cells, known as cribriform pattern 4 (CP4), which can indicate how aggressive a cancer might be. By studying patients who had undergone surgery to remove the prostate gland, we found that those with CP4 tended to have a higher risk of cancer recurrence. Additionally, we discovered certain genes that behaved differently in patients with CP4, suggesting unique genetic changes associated with this pattern. Our findings suggest that patients with CP4 might need more intense treatment after surgery and could help doctors decide on the best course of action. Importantly, our research provides new insights that could lead to better management of prostate cancer, aiming to improve patient outcomes.

**Abstract:**

Purpose: Given the diverse clinical progression of prostate cancer (PC) and the evolving significance of histopathological factors in its management, this study aimed to explore the impact of cribriform pattern 4 (CP4) on clinical outcomes in PC patients and examine its molecular characteristics. Methods: This retrospective study analyzed data from The Cancer Genome Atlas (TCGA) database and included PC patients who underwent radical prostatectomy (RP) and had pathology slides available for the assessment of CP4. A multivariable competing risk regression analysis was used to assess the association between CP4 and progression-free survival (PFS) while adjusting for established PC prognostic factors. The frequency of genomic alterations was compared between patients with and without CP4 using the Fisher’s exact test. Results: Among the 394 patients analyzed, 129 (32.74%) had CP4. After a median follow-up of 40.50 months (IQR: 23.90, 65.60), the presence of CP4 was significantly associated with lower PFS (AHR, 1.84; 95% CI, 1.08 to 3.114; *p* = 0.023) after adjusting for covariates. Seven hub genes—KRT13, KRT5, KRT15, COL17A1, KRT14, KRT16, and TP63—had significantly lower mRNA expression levels in patients with CP4 compared to those without. Conclusions: PC patients with CP4 have distinct genomic alterations and are at a high risk of disease progression following RP. Therefore, these patients may benefit from additional post-RP treatments and should be the subject of a prospective randomized clinical trial.

## 1. Introduction

Risk stratification plays a critical role in the management of prostate cancer (PC) as it directly impacts treatment strategies. A key element in this process is histopathology, which has seen significant evolution over the years. The Gleason score, first introduced in 1977, remains a cornerstone in this domain, though its definitions and criteria have been continuously refined [1]. The most recent update to the Gleason scoring system mandates that all cribriform glands be classified as Gleason pattern 4 [2]. Consequently, the clinical implications of cribriform pattern 4 (CP4) in PC have gained a growing interest in recent years, particularly due to its association with adverse clinical outcomes [3,4,5,6,7,8,9,10]. A comprehensive meta-analysis highlighted these concerns, revealing that the presence of CP4 is associated with increased risks for biochemical recurrence (hazard ratio [HR]: 2.14; *p* < 0.01) and cancer-specific mortality (HR: 3.30; *p* < 0.01), as well as increased likelihood of extra-prostatic extension (odds ratio [OR] 1.96; *p* < 0.0001), seminal vesicle invasion (OR: 2.89; *p* < 0.01), and positive surgical margins (OR: 1.88; *p* < 0.0007) [11]. These findings emphasize the critical role of CP4 in prognosticating the course of PC and highlight the need for continued research in this area.

The current standard of care for patients with PC following radical prostatectomy (RP) involves salvage radiation therapy (RT) when the prostate-specific antigen (PSA) level reaches 0.1 ng/mL. Identifying specific growth patterns such as CP4 is becoming increasingly crucial, as they hold potential to refine post-prostatectomy treatment strategies. While the clinical relevance of CP4 in PC is evolving, the understanding of its molecular characteristics remains limited. In this study, we analyze the impact of CP4 on progression-free survival (PFS), adjusting for recognized PC prognostic factors, and explore CP4 molecular characteristics. This research aims to identify patients who may benefit from adjuvant therapies, thereby guiding the development of future randomized clinical trials.

## 2. Materials and Methods

### 2.1. Patient Characteristics

This retrospective study involved an analysis of patient data from The Cancer Genome Atlas (TCGA) database. The selection criteria included patients diagnosed with PC, who had undergone RP and had digitized hematoxylin-and-eosin (HE)-stained slides with clearly identifiable tumor sections available for review. Demographic and clinical data, including age at diagnosis, pathologic tumor stage, margin status, Gleason score, and pre-RP PSA levels, were collected due to their established significance in predicting PC prognosis and treatment outcomes. The follow-up period was defined from the date of RP to the last available follow-up or death.

A pathological review of the RP specimens was undertaken by a team of three pathologists with experience in genitourinary pathology (MA, ET, SK). The primary objective was to identify cases with CP4. This entailed a thorough review of the pathology slides, with a special emphasis on identifying the presence and characteristics of cribriform architectural structures. Distinguishing between CP4 and intraductal carcinoma in our study was primarily based on the difference in their morphologic features, as described in previous literature [12,13].

### 2.2. Statistical Methods

#### 2.2.1. Comparison of the Distribution of Clinical Factors Stratified by Cribriform Pattern 4

This study followed a prespecified statistical analysis plan. Descriptive statistics provided an overview of the clinical and treatment characteristics of 394 patients, categorized based on whether they exhibited CP4 or not. The comparison of these characteristics utilized the Pearson’s chi-squared test for categorical variables and the Wilcoxon two-sample test for analyzing distribution differences in continuous variables, such as age and PSA levels [14]. Additionally, the distribution of the follow-up times was assessed using the reverse Kaplan–Meier method and evaluated for significance with a log-rank *p*-value [15].

#### 2.2.2. Covariate-Adjusted Hazard Ratios and Estimates of Progression-Free Survival

The primary endpoint of this study was PFS, defined as the time from RP to the first evidence of disease progression or death from any cause [16,17,18]. Univariable and multivariable competing risk regression analysis using the Fine–Gray mode was applied to assess whether a significant association existed between the presence of CP4 and PFS [19]. The covariates included in the model were age at the time of diagnosis, pre-RP PSA, pathologic tumor stage, Gleason score, and margin status. Time 0 was the date of RP. For the purpose of illustration, covariate-adjusted estimates of PFS following RP, stratified by the presence or absence of CP4, were calculated using the Kaplan–Meier method [20]. The *p*-values for the adjusted Kaplan–Meier plots were calculated using the log-rank test. The threshold for statistical significance was established at *p* < 0.05. All statistical analyses were performed with R (version 4.2.3).

#### 2.2.3. Molecular Characterization and Genomic Alterations Analysis

The expression levels of mRNAs from the TCGA dataset were examined, and genes meeting the criteria of an absolute log fold change (|logFC|) > 1 or <−1, along with a q-value < 0.05, were identified as differentially expressed genes (DEGs) in patients with and without CP4 [21].

Following the identification of DEGs, a protein–protein interaction (PPI) network was constructed using the STRING tool [22], and the results were visualized using Cytoscape software v310.1 [23]. A topological analysis was performed on each gene to identify hub genes, which are genes highly interconnected within the PPI network. Subsequently, Molecular Complex Detection (MCODE) plugin was applied to identify significant modules within the PPI network, using module detection criteria that included a node score cutoff of 0.2, a K-core value of 2, and a degree cutoff of 2 [24]. As a result, seven hub genes (KRT13, KRT5, KRT15, COL17A1, KRT14, KRT16, TP63) were identified. Subsequently, the frequency of genomic alterations was compared between patients with and without a CP4 using the Fisher’s exact test.

## 3. Results

### 3.1. Comparison of the Distribution of Clinical Factors Stratified by Cribriform Pattern 4

Among the 394 patients analyzed, 129 (32.74%) had CP4. The characteristics of these patients, stratified by the presence of CP4, are presented in Table 1. Compared to those without CP4, the patients with CP4 had a longer median follow-up [46.37 months (IQR: 24.13, 76.97) versus 37.27 months (IQR: 22.87, 62.90), *p* = 0.001], a higher pre-RP PSA [8.10 ng/mL (IQR: 5.20, 14.40) versus 7.00 ng/mL (IQR: 5.00, 9.9), *p* = 0.007], and a higher Gleason score (52% versus 27%, *p* = 0.001) and were more likely to have T3a or higher stage PC (76% versus 53%, *p* < 0.001).

### 3.2. Covariate-Adjusted Hazard Ratios and Estimates of Progression-Free Survival

After a median follow-up of 40.50 months (IQR: 23.90, 65.60), the outcomes included biochemical recurrence in 79 men (20.05%), locoregional recurrence in 5 (1.26%), distant metastasis in 2 (0.50%), new primary tumors in 2 (0.50%), and 4 deaths (1.01%). As shown in Table 2, the presence of CP4 was associated with a significantly lower PFS (AHR, 1.84; 95% CI, 1.08 to 3.11; *p* = 0.023) after adjusting for known covariates. As illustrated in Figure 1, men with CP4 had significantly lower estimates of adjusted PFS compared to those without it (*p* < 0.001), with the adjusted 10-year PFS estimate being 85.72% (95% CI, 74.79% to 98.24%) for men with CP4 versus 93.01% (95% CI, 87.79% to 98.54%) for those without.

### 3.3. Genetic and Molecular Profiles Stratified by the Cribriform Pattern

The expression levels of two specific mRNAs were significantly elevated in patients with CP4, while 90 other mRNAs exhibited significantly reduced expression levels in these patients compared to those without CP4 (Figure 2a). Furthermore, seven hub genes, namely, KRT13, KRT5, KRT15, COL17A1, KRT14, KRT16, and TP63, exhibited significantly lower mRNA expression levels in patients with CP4 compared to those without (Figure 2b–h).

The FRG1BP gene displayed the highest mutation frequency in both groups, with 50 mutations affecting 26.1% of the patients with CP4 and 69 mutations affecting 16.6% of the patients without CP4. Among the hub genes, only genomic alterations in COL17A1 significantly differed between patients with CP4 and those without (Figure 3).

## 4. Discussion

In this study, we found that the presence of CP4 in PC patients undergoing RP was associated with reduced PFS, even when accounting for known PC prognostic factors, and was also linked to distinct genomic alterations. The clinical significance of these findings is that they allow for identifying a subset of patients with adverse histopathological characteristics who are at a higher risk of reduced PFS and have unique genomic alterations. This information is valuable for the selection of candidates for future prospective randomized clinical trials designed to explore additional treatments following RP.

In recent years, specific morphologic features, such as CP4, have been increasingly recognized as significant predictors of prognosis in PC patients, highlighting their importance in the stratification of disease aggressiveness. Oufattole et al. highlighted that the presence of CP4 or comedo/solid components in Gleason score 9–10 PC is associated with higher rates of lymphovascular invasion and biochemical recurrence, suggesting that the CP4 morphology is an independent predictor of a higher risk of biochemical recurrence [25]. Similarly, Okubo et al. demonstrated that the presence of CP4 in prostate biopsy specimens is an independent risk factor for lymph node metastasis, showing the prognostic significance of this morphologic feature even when partially present [26]. Moreover, Shimodaira et al. revealed that in patients with Gleason score 8 PC, the area and percentage of CP4 are independent predictors of biochemical recurrence after robot-assisted radical prostatectomy [27]. These findings collectively demonstrate the prognostic significance of CP4 in prostate cancer, reinforcing its role in guiding post-surgical management and highlighting the need for a meticulous histopathological evaluation to identify CP4. Furthermore, the study by Di Mauro et al. showed the critical role of pre-operative assessments, revealing that the PIRADS scores at multiparametric MRI are significant predictors of CP4 in patients undergoing RP, thereby reinforcing the necessity of incorporating advanced imaging findings to enhance the prognostic accuracy for PC patients with CP4 [28]. Our analysis supports these conclusions, demonstrating the association of CP4 with reduced PFS, thereby emphasizing the value of including this morphologic feature in the prognostic assessment of PC patients.

The current standard of care for patients who have undergone RP involves initiating salvage RT once the PSA level reaches 0.1 ng/mL. This approach is supported by a meta-analysis of three prospective randomized controlled trials, focusing on PFS as the primary endpoint [29]. However, concerns have been raised regarding the potential impact of immortal time bias on the results of the meta-analysis, suggesting that a subgroup of high-risk patients might benefit from adjuvant RT [30]. Supporting this, evidence from a large multi-national and multi-institutional study indicated a significant reduction in all-cause mortality with the use of adjuvant RT over salvage RT, particularly in patients with a pathological Gleason score of 8–10 and stage-pT3 or -pT4 PC [31]. Furthermore, the ongoing NRG-GU008 randomized control trial is exploring treatment intensification with abiraterone and apalutamide following RP in patients with pathologically node-positive disease and detectable PSA [32]. Patients with CP4 post-RP may also benefit from adjuvant RT or treatment intensification, highlighting the need for their inclusion in future randomized trials.

Our findings contribute to the limited but growing body of literature on the molecular characteristics of PC with CP4. Notably, the distinct molecular features observed in our study, such as the differential expression and mutation frequency of genes like FRG1BP and alterations in COL17A1, align with and extend previous research by Elfandy et al., who reported unique genetic, transcriptional, and epigenetic features distinguishing CP4 from non-cribriform pattern 4 (non-CP4) tumors [33]. These features include increased somatic copy number variations, mutations in SPOP and ATM, and pathway enrichments, which collectively suggest a distinct molecular phenotype for CP4 PC that closely resembles that of metastatic PC and is associated with progression to lethal disease [33]. Similarly, our study echoes the findings of Böttcher et al., who reported that CP4 is linked to increased genomic instability, highlighting CP4 role as a pathological substrate for progressive molecular tumor derangement [34]. Together, these findings emphasize CP4 significance in the molecular landscape of PC, highlighting its potential as a marker of aggressive disease and a target for future therapeutic strategies.

Several points require further discussion. First, histologic interpretations in this study were based on one or two virtual slides. Considering the pathology review committee’s limited access to the entire prostatectomy specimens, it is possible that patients classified as lacking evidence of cribriform pattern 4 could, in fact, present this pattern in unreviewed sections of the specimens. Despite this limitation, a significant association was observed between the presence of CP4 and reduced PFS. Second, it is important to emphasize that although men with CP4 are more likely to present with higher pre-RP PSA levels and Gleason scores and advanced-stage disease (T3a or higher), these variables were carefully considered and adjusted for in the multivariable model. Third, the TCGA database lacks detailed longitudinal PSA measurements and imaging results post-RP, necessitating our PFS assessment to be based solely on available dates of diagnosis, treatment, and significant clinical events. Fourth, the absence of data on adjuvant treatments in our dataset represents a limitation, potentially impacting the interpretation of the PFS outcomes. Finally, as this is a retrospective study, the results are hypothesis-generating and warrant further investigation in a prospective cohort study.

## 5. Conclusions

The association of CP4 with reduced PFS in PC patients undergoing RP highlights the importance of identifying patients at increased risk of disease progression. These findings advocate for the potential benefits of adjuvant treatment in this subgroup, emphasizing the need for further investigation through prospective randomized clinical trials. Moreover, the distinct genetic and molecular characteristics of CP4 further solidify its role as a critical marker of aggressive PC, reinforcing the importance of targeted treatment strategies and comprehensive molecular profiling in future clinical trials.

## Figures and Tables

**Figure 1 cancers-16-01248-f001:**
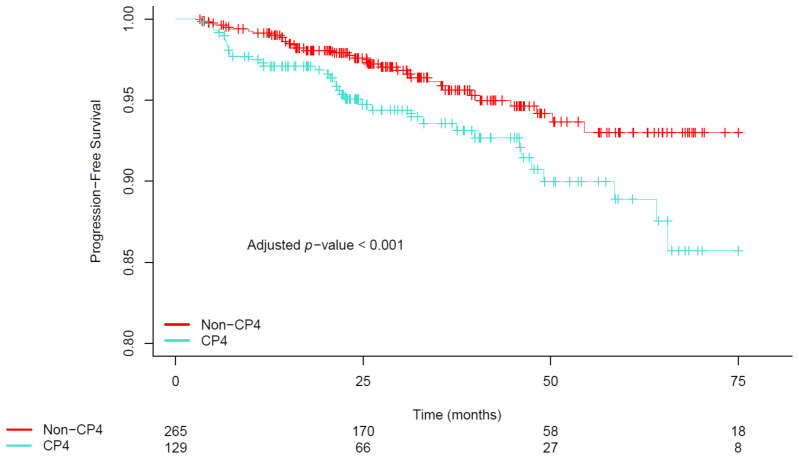
Covariate-adjusted estimates of progression-free survival stratified by the presence of cribriform pattern 4. Abbreviations: CP4, cribriform pattern 4; non-CP4, non-cribriform pattern 4.

**Figure 2 cancers-16-01248-f002:**
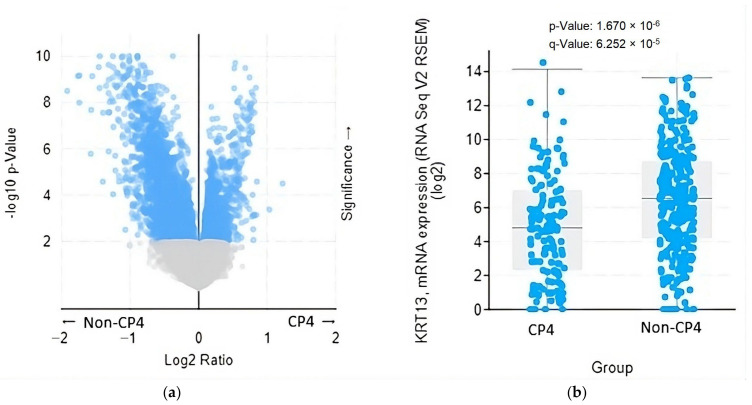
The mRNA expression level of (**a**) the differentially expressed genes (DEGs) and (**b**–**h**) the seven hub genes stratified by the presence of cribriform pattern 4. Abbreviations: CP4, cribriform pattern 4; non-CP4, non-cribriform pattern 4.

**Figure 3 cancers-16-01248-f003:**
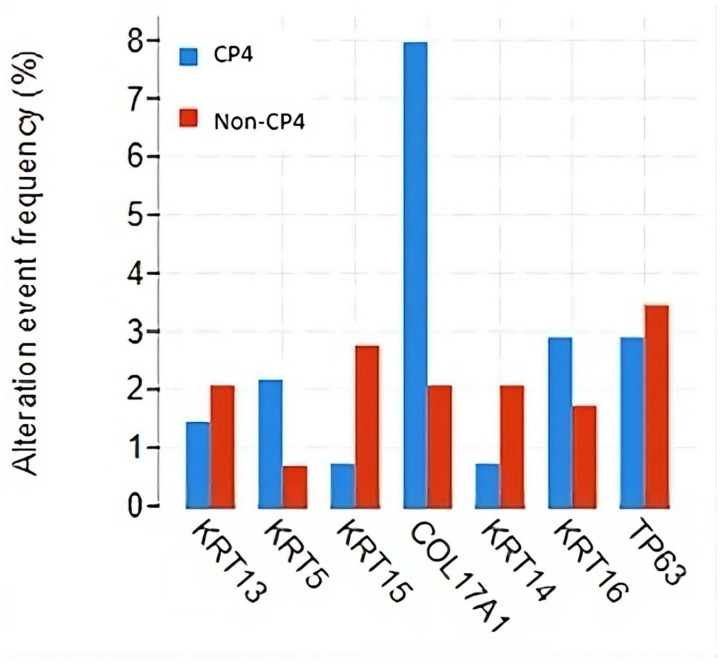
Genomic alterations in the seven hub genes stratified by the presence of cribriform pattern 4. Abbreviations: CP4, cribriform pattern 4; non-CP4, non-cribriform pattern 4.

**Table 1 cancers-16-01248-t001:** Comparison of the distribution of clinical and treatment factors stratified by the presence of the cribriform pattern.

	CP4 (*n* = 129)	Non-CP4 (*n* = 265)	*p*
Age (years), median (IQR)	62 (57, 66)	61 (55, 66)	0.174
Pre-RP PSA, ng/mL, median (IQR)	8.10 (5.20, 14.40)	7.00 (5.00, 9.9)	0.007
Prostatectomy tumor stage, No. (%)			<0.001
T2	30 (24%)	124 (47%)	
T3a or higher	97 (76%)	138 (53%)	
Prostatectomy Gleason score, No. (%)			<0.001
7 or less	60 (48%)	188 (73%)	
8–10	66 (52%)	69 (27%)	
Prostatectomy margin status, No. (%)			0.200
Negative	92 (72%)	167 (66%)	
Positive	35 (28%)	87 (34%)	
Prostatectomy nodal status, No. (%)			0.300
Negative (N0)	99 (81%)	181 (86%)	
Positive (N1)	23 (19%)	30 (14%)	
* Follow-up (month), median (IQR)	46.37 (24.13, 76.97)	37.27 (22.87, 62.90)	0.001

Abbreviations: CP4, cribriform pattern 4; non-CP4, non-cribriform pattern 4; RP, radical prostatectomy; PSA, prostate-specific antigen; IQR, interquartile range.* reverse KM method is used to estimate time of median follow-up.

**Table 2 cancers-16-01248-t002:** Covariate-adjusted hazard ratios for progression-free survival.

Covariates	Univariable			Multivariable		
	HR	95% CI	*p*	AHR	95% CI	*p*
Age (years)	0.993	0.967–1.019	0.601	0.977	0.950–1.005	0.109
Baseline PSA, ng/mL						
<4	0.990	0.409–2.398	0.984	0.616	0.233–1.626	0.327
4–10	Reference	Reference		Reference	Reference	
>10	1.463	0.917–2.335	0.110	0.741	0.431–1.275	0.279
Prostatectomy tumor stage, No.						
T2	Reference	Reference		Reference	Reference	
T3a or higher	4.949	2.450–9.999	<0.001	3.903	1.693–8.995	0.001
Prostatectomy Gleason score						
7 or less	Reference	Reference		Reference	Reference	
8–10	3.461	2.198–5.472	<0.001	2.279	1.342–3.870	0.002
Prostatectomy margin status						
Negative	Reference	Reference		Reference	Reference	
Positive	1.604	1.017–2.530	0.042	1.117	0.642–1.945	0.693
Cribriform pattern 4						
Non-CP4	Reference	Reference		Reference	Reference	
CP4	2.210	1.429–3.417	<0.001	1.837	1.080–3.114	0.023

Abbreviations: HR, hazard ratio; CI, confidence interval; AHR, adjusted hazard ratio; CP4, cribriform pattern 4; non-CP4, non-cribriform pattern 4.

## Data Availability

The original contributions presented in the study are included in the article. Further inquiries can be directed to the corresponding author.

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
