# Peer review of "Prognostic Significance of the Cribriform Pattern in Prostate Cancer: Clinical Outcomes and Genomic Alterations"

_cancers, 2024, doi:10.3390/cancers16071248_

Round 1

Reviewer 1 Report

Comments and Suggestions for Authors

In this interesting retrospective study, the authors analyzed the prognostic role of CP4 in prostate cancer and studied its molecular characteristics. Their results provide further evidence that CP4 represents an unfavorable prognostic parameter in patients with PC and that this pattern has a characteristic genomic profile.

The study design is appropriate in relation to the research question, the methods are clearly described, and the conclusions are supported by the findings reported. The article is well structured and fluent, and all the references are proper and strictly relevant to the study.

I have only a few comments:

- The authors are encouraged to discuss the prognostic role of CP4 in light of more recent studies, such as 10.1111/his.14901, 10.1016/j.humpath.2023.01.008, 10.1002/cam4.7086, and so on

- The authors acknowledge that the assessment of cribriform histology on 1-2 (H&E?) virtual slides is a limitation of their study. How did they address the challenging issue of distinguishing between GP4 and intraductal carcinoma? Please make a statement on this

Author Response

Comment 1: “The authors are encouraged to discuss the prognostic role of CP4 in light of more recent studies, such as 10.1111/his.14901, 10.1016/j.humpath.2023.01.008, 10.1002/cam4.7086, and so on”

Response 1: We sincerely appreciate the reviewer’s insightful suggestions and the opportunity to enhance our manuscript by discussing the prognostic role of CP4 in prostate cancer in light of more recent studies. As recommended, we have included a new paragraph in the discussion section of our manuscript (Page 8, Lines 185-205). This paragraph specifically addresses the recent findings on the prognostic significance of CP4 in prostate cancer.

Comment 2: “The authors acknowledge that the assessment of cribriform histology on 1-2 (H&E?) virtual slides is a limitation of their study. How did they address the challenging issue of distinguishing between GP4 and intraductal carcinoma? Please make a statement on this.”

Response 2: We thank the reviewer for highlighting an important point. TCGA dataset only includes H&E stained slides of cases and lack of supplementary biomarker studies, such as the biomarkers expressed by basal cells, limit the determination of whether the complex cribriform architecture is invasive cribriform or intraductal carcinoma. However, current evidence (e.g. PMID: 16980940; 29297491) give us adequate morphologic features of intraductal carcinoma, including presence of basal cell layer (mostly appreciated with H&E stained slides; cribriform pattern 4 lacks basal cell layer), dense solid or packed cribriform growth with few lumens (in contrast,  cribriform growth pattern has more lumens); and frequent bizarre cytologic atypia (not a diagnostic criterion for intraductal carcinoma of the prostate but regardless most of them have it; cribriform pattern 4 prostatic adenocarcinoma has usual monomorphic histology). Therefore, while having H&E slides only is a limitation for definitive determination of intraductal carcinoma, expert urologic pathologists can diagnose intraductal carcinoma without additional biomarker assays.  We have now included a statement in our manuscript to clarify how we distinguished the CP4 and intraductal carcinoma (Page 2, Lines 85-87)

Reviewer 2 Report

Comments and Suggestions for Authors

The authors should be congratulated for their work. The manuscript aimed to analyze the impact of the cribriform pattern 4 (CP4) on progression-free survival (PFS), and to explore its molecular characteristics, in prostate cancer within TGCA database.. The results observed that the presence of CP4 was associated with lower PFS than the absence of CP4. These results are in line with the current literature, however, the manuscript is very difficult to follow due to the several lackings:

- The PFS should be defined as follows: "the length of time during and after the treatment of a disease, such as cancer, that a patient lives with the disease but it does not get worse. In a clinical trial, measuring progression-free survival is one way to see how well a new treatment works." (https://www.cancer.gov/publications/dictionaries/cancer-terms/def/progression-free-survival). As the authors want to keep their definition, a reference is needed. Moreover, how was measured the PFS? Imaging? PSA tests? Which were the PSA values considered? 

- The multivariable model is misunderstandable. The multivariable model should address the mortality. Moreover, it is wrong and also counterintuitive to say that Gleason 8-10 patients had higher PFS. They exhibited lower PFS due to their cancer aggressiveness. The same consideration should apply to the other covariates (PSA, T stage, and margin status). Moreover, the non-CP4 should be put on top (it is the variable of interest) and should be the reference if the authors addressed the impact of CP4 on PFS. 

- The Kaplan Meyer should be stopped at 75 months. No sense to show 150 months with 1 person involved. 

- The cribriform pattern is involved also in the upgrading of prostate cancer. Several novel papers addressed this endpoint and should be discussed (PMID= 35838831, 36984626).

Is any data available on MRI features? PIRADS? It would be interesting to know if higher PIRADS is related to the presence of CP4. 

Is any data available on the adjuvant treatment of these patients? RT or hormone therapy? and on Lymph node status? All these are inherent limitations.

Author Response

Comment 1: “The PFS should be defined as follows: "the length of time during and after the treatment of a disease, such as cancer, that a patient lives with the disease but it does not get worse. In a clinical trial, measuring progression-free survival is one way to see how well a new treatment works." (https://www.cancer.gov/publications/dictionaries/cancer-terms/def/progression-free-survival). As the authors want to keep their definition, a reference is needed. Moreover, how was measured the PFS? Imaging? PSA tests? Which were the PSA values considered?”

Response 1: We greatly appreciate the reviewer’s insightful feedback regarding the definition of PFS utilized in our manuscript. The reviewer’s comment has prompted us to revisit our definition and the rationale behind its use in the context of our study. Our definition of PFS as "the duration from radical prostatectomy (RP) to the first evidence of disease progression or death from any cause" was chosen to reflect both the clinical relevance of disease progression in prostate cancer research and the specific endpoints relevant to the patient cohort under study. This definition aligns with the definitions used in several seminal studies in the field (PMID: 18421050, 15647217, 19903805). In response to the reviewer’s valuable suggestion, we have now included these references within our manuscript to provide a robust foundation for our definition of PFS (Page 3, Line 100).

Our study utilized patient data from the TCGA database, which, while providing a rich source of genomic and clinical information, unfortunately does not include detailed longitudinal data on PSA levels or subsequent imaging results post-radical prostatectomy. However, it does provide the dates of significant events such as biochemical failure, locoregional recurrence, death, etc. Consequently, our assessment of PFS was based on the clinical data presented within the TCGA dataset, which includes dates of diagnosis, treatment, and last follow-up or events (progression or death). In the clinical setting, progression in prostate cancer is often first indicated by rising PSA levels, with further investigations through imaging recommended based on specific PSA thresholds, commonly when PSA reaches 0.2 ng/mL or more after radical prostatectomy. This approach, however, could not be directly applied in our analysis due to the nature of the TCGA data. We have now include this under the limitations of the manuscript.

Comment 2: “The multivariable model is misunderstandable. The multivariable model should address the mortality. Moreover, it is wrong and also counterintuitive to say that Gleason 8-10 patients had higher PFS. They exhibited lower PFS due to their cancer aggressiveness. The same consideration should apply to the other covariates (PSA, T stage, and margin status). Moreover, the non-CP4 should be put on top (it is the variable of interest) and should be the reference if the authors addressed the impact of CP4 on PFS.”

Response 2: We thank the reviewer for their comment regarding the incorporation of mortality in our multivariable model. The Fine and Gray model is specifically designed to handle competing risks by considering them as separate outcomes that can alter the probability of the event of interest (i.e., disease progression) occurring. In our study, PFS was defined as the duration from radical prostatectomy to the first evidence of disease progression or death from any cause, aligning with the Fine and Gray competing risks regression model used for our analysis. In implementing the Fine and Gray model using R, we required an "event" variable coded as "0-1" and a "time" variable. Consistent with R's conventions (where 0 typically represents "alive" and 1 represents "dead"), we adapted our dataset accordingly. In our case, the "event" variable represents not just mortality but any defined progression event, including disease recurrence or death. This was operationalized by coding the “days_to_new_tumor_event_after_initial_treatment" field from our database as 1 if an event (disease progression or death) occurred and 0 if no event was observed within the study period. This approach ensures that our multivariable model adequately addresses mortality by treating it as a component of the composite outcome variable, thereby capturing the full spectrum of clinical outcomes relevant to PFS in our analysis.

We agree with the reviewer assessment that patients with Gleason scores of 8-10, indicative of more aggressive cancer, indeed tend to have lower PFS. This relationship is accurately reflected in our results and specifically detailed in Table 2 of our manuscript (Page 4-5, Line 148). Specifically, our analysis demonstrates that patients with a prostatectomy Gleason score of 8-10 are 2.279 times more likely to experience a progression event compared to patients with a prostatectomy Gleason score of 7 or less (P = 0.002), signifying a significant association between higher Gleason scores and reduced PFS.

The placement of non-CP4 and CP4 in Table 2 was indeed a typographical error during the table generation process. We concur with the reviewer’s recommendation that non-CP4 should be listed as the reference category and thus be positioned at the top, given our focus on assessing the impact of CP4 on PFS. This error has been corrected in the revised version of Table 2, ensuring that non-CP4 is now appropriately placed at the top as the reference, and CP4 is listed subsequently (Page 4-5, Line 148). We appreciate the reviewer’s keen observation, which has helped us improve the accuracy and readability of our manuscript.

Comment 3: “The Kaplan Meyer should be stopped at 75 months. No sense to show 150 months with 1 person involved.”

Response 3: We appreciate the reviewer’s recommendation regarding the presentation of the Kaplan-Meier survival curves. Following their suggestion, we have updated Figure 1 to limit the follow-up duration to 75 months (Page 4, Line 145). This change addresses the concern of extending the analysis to 150 months with a minimal number of patients contributing to the data at that point, ensuring a more meaningful and interpretable presentation of the outcomes.

Comment 4: “The cribriform pattern is involved also in the upgrading of prostate cancer. Several novel papers addressed this endpoint and should be discussed (PMID= 35838831, 36984626).”

Response 4: We appreciate the reviewer’s suggestions and the opportunity to enhance our manuscript by discussing the additional aspects of the CP4 in prostate cancer.  As also recommended by another reviewer, we have included a new paragraph in the discussion section of our manuscript (Page 8, Lines 185-205).

Comment 5: “Is any data available on MRI features? PIRADS? It would be interesting to know if higher PIRADS is related to the presence of CP4.”

Response 5: We thank the reviewer’s comment regarding the inclusion of MRI features and PIRADS scores in our study. Unfortunately, this information was not included the TCGA database.

Comment 6: “Is any data available on the adjuvant treatment of these patients? RT or hormone therapy? and on Lymph node status? All these are inherent limitations.”

Response 6: We appreciate the reviewer’s inquiry regarding the availability of data on adjuvant treatments and lymph node status for the patients in our study. Lymph node status has been included and is presented in Table 1 of our manuscript, providing important pathological details relevant to our study cohort (Pages 3-4, Lines 134-135). However, we must acknowledge that our dataset, sourced from the TCGA database, does not contain detailed records on the adjuvant treatments. We have now updated the limitations section of our manuscript to explicitly mention this.

Reviewer 3 Report

Comments and Suggestions for Authors

 The presented manuscript shows the work conducted by a large group of co-authors, who have started to write this manuscript given the diverse clinical progression of prostate cancer (PC) and the evolving significance of histopathological factors in its management. Interestingly, this study aims to explore the impact of the cribriform pattern 4 (CP4) on clinical outcomes in PC patients and examine its molecular characteristics. Anyway, it represents a retrospective study analyzed data from The Cancer Genome Atlas (TCGA) database and included PC patients who underwent radical prostatectomy (RP) and had pathology slides available for the assessment of the CP4. In conclusion, the authors wrote that the PC patients with a CP4 have a distinct genomic alternation and are at a high risk of disease progression following RP and consequently these patients may benefit from additional post-RP treatments and should be the subject of a prospective randomized clinical trial. Generally, it is a well-structured paper and the used citations correspond to updated references, though it needs several serious improvements, like the quality of the graphics is not sufficient for being published in a journal. You should increase the quality of the images. In addition, I recommend the authors extend the Discussion section because there are just a few referenced results included in the comparison of the authors' results with the previous scientific works. I would highly recommend doing the suggested modifications.

Author Response

We thank the reviewer for their positive feedback and for dedicating their time to review our manuscript. We have addressed the specific comments below.

Comment 1: “…the quality of the graphics is not sufficient for being published in a journal. You should increase the quality of the images.”

Response 2: We thank the reviewer for their recommendation. The quality of the graphics has now been further improved in the figures.

Comment 2: “In addition, I recommend the authors extend the Discussion section because there are just a few referenced results included in the comparison of the authors' results with the previous scientific works.”

Response 2: We appreciate the reviewer's suggestion and have accordingly expanded the discussion section with an additional paragraph (Page 8, Lines 185-2005) and have further elaborated on the study's limitations (Page 9, Lines 234-248).

Round 2

Reviewer 2 Report

Comments and Suggestions for Authors

The authors had properly addressed my suggestions. Now the manuscript is fit for publication.